# Deciphering the *Monilinia fructicola* Genome to Discover Effector Genes Possibly Involved in Virulence

**DOI:** 10.3390/genes12040568

**Published:** 2021-04-14

**Authors:** Laura Vilanova, Claudio A. Valero-Jiménez, Jan A.L. van Kan

**Affiliations:** 1Laboratory of Phytopathology, Wageningen University, 6708 PB Wageningen, The Netherlands; laura.vilanova@irta.cat (L.V.); claudiovalero@gmail.com (C.A.V.-J.); 2IRTA, Postharvest Programme, Edifici Fruitcentre, Parc Científic i Tecnològic Agroalimentari de Lleida, Parc de Gardeny, 25003 Lleida, Catalonia, Spain

**Keywords:** brown rot, stone fruit, annotation, necrosis, cell-death

## Abstract

Brown rot is the most economically important fungal disease of stone fruits and is primarily caused by *Monilinia laxa* and *Monlinia fructicola*. Both species co-occur in European orchards although *M. fructicola* is considered to cause the most severe yield losses in stone fruit. This study aimed to generate a high-quality genome of *M. fructicola* and to exploit it to identify genes that may contribute to pathogen virulence. PacBio sequencing technology was used to assemble the genome of *M. fructicola*. Manual structural curation of gene models, supported by RNA-Seq, and functional annotation of the proteome yielded 10,086 trustworthy gene models. The genome was examined for the presence of genes that encode secreted proteins and more specifically effector proteins. A set of 134 putative effectors was defined. Several effector genes were cloned into *Agrobacterium tumefaciens* for transient expression in *Nicotiana benthamiana* plants, and some of them triggered necrotic lesions. Studying effectors and their biological properties will help to better understand the interaction between *M. fructicola* and its stone fruit host plants.

## 1. Introduction

Brown rot is an economically important fungal disease that mainly affects stone fruit such as peaches, nectarines, cherries and apricots and nowadays is also causing important problems in almond [1]. This disease affects blossom, twig, and ripe as well as unripe fruit during pre- or post-harvest handling. Mummified fruit that remains on trees or the orchard floor during the winter generate the inoculum for newly emerging flowers and fruit in the subsequent spring. During unfavourable environmental conditions and/or in unripe fruits, primary infections can remain latent until favourable conditions develop that facilitate disease progression and fruit decay. Moreover, decayed fruit can serve as a source for secondary infection during the summer season. Despite the infection occurring in the field, the main economic losses of brown rot appear during postharvest handling, storage and transport.

Several fungal species are responsible of stone fruit brown rot disease including *Monilinia fructicola*, *M. laxa* and *M. fructigena*. *M. fructigena* is more common in pome fruit and has lower incidence in stone fruit. Like *M. fructigena*, *M. laxa* is endemic in Europe and is a quarantine pathogen in China and the United States. In contrast, *M. fructicola* is endemic in North America, Australia and Japan and was a quarantine pathogen in Europe for several years [2] until it was spread across the continent [3]. Nowadays, all three species co-occur in European orchards although *M. fructicola* is considered to cause the most severe yield losses in stone fruit [4].

The control of the disease is mainly attempted in the field since there are only a few registered postharvest fungicides available to control *Monilinia* spp. A key step is the application of fungicides during blossom blight to reduce fruit decay and thereby prevent primary infections. The public concern about health problems, the impact on the environment and the appearance of fungicide-resistant fungal strains have stimulated research to develop new methods to control brown rot in stone fruits. Some of the alternatives focused on the use of physical methods and the application of biological control agents [5,6,7]. Despite all the efforts deployed to find new strategies to control *Monilinia* spp. none of them were able to replace fungicides until now. 

The genus *Monilinia* belongs to the family *Sclerotiniaceae* and comprises ~25 species, the majority of which are plant pathogens with a necrotrophic lifestyle. Fungi with a necrotrophic lifestyle were initially considered to act as brutal killers who degrade plant tissue with the secretion of a battery of cell wall-degrading enzymes and toxins in an indiscriminate manner [8]. Lately, however, it was revealed that many necrotrophic fungi actively manipulate the programmed cell death machinery of the host for their own benefit by secreting effector proteins and/or secondary metabolites [9,10]. There are several well-studied effectors secreted by necrotrophic pathogens infecting wheat like *Parastagonospora nodorum* and *Pyrenophora tritici-repentis. P. nodorum* uses effectors to induce plant programmed cell death and facilitate the infection of hosts that contain susceptibility genes [11,12,13,14]. In the case of *P. tritici-repentis,* Tan et al. [15] reported that the effector ToxA had a major impact on cultivar choice and breeding strategies. In many cases, plant sensitivity to necrotrophic effectors is determined by a dominant gene, and disease does not occur in the absence of effector sensitivity [11,13,14]. Until now, the research on necrotrophic effectors was mainly focused on fungal pathogens affecting extensive, arable crops such as wheat, barley and maize. In the case of *Botrytis cinerea* (the most intensely studied species in the *Sclerotiniaceae*), several cell death-inducing effector molecules (proteins or metabolites) were identified, however, none of them were essential for virulence [16,17,18]. The genus *Monilinia* is characterized to less extent because both the fungi and their hosts have long been less amenable to molecular-genetic studies. There are only a few reports on *Monilinia* spp. virulence factors [19,20] but none focusing on the role of effector proteins inducing cell death in stone fruit leaves or fruits. 

The advent of high throughput “omics” technologies such as mRNA sequencing (RNA-Seq), has provided unprecedented access to the fine structure of the transcriptome. The analyses of fungal pathogen genome and transcriptome sequences offer options to perform an identification and functional analysis of substances that are specifically toxic to their respective hosts, and thereby confer on the fungi their host specificity. In recent years, many fungal genomes were analysed mainly because of the decrease in sequencing costs. In the case of plant pathogenic fungi, many species have been sequenced to establish evolutionary lineages and to focus on the virulence factors that participate in the plant infection process [21]. The genome sequences of different *M. fructicola* strains [22,23] as well as the *M. fructicola* mitochondrial DNA sequence [24] were recently published however, none of these were yet used to get new insights into the pathogen biology or to study the interaction of *M. fructicola* with host plants.

The aim of this study was to generate a high-quality genome of *M. fructicola* using PacBio sequencing and perform a manual structural curation from the annotated protein-coding genes. In genome curation, we paid specific attention to genes that may contribute to pathogen virulence, such as genes encoding secreted plant cell wall-degrading enzymes and genes encoding effector proteins. Identifying protein effectors and studying their biological functions will help to better understand the interaction between necrotrophic fungi and host plants.

## 2. Materials and Methods

### 2.1. Strains and Culture Conditions

The strain *M. fructicola* CPMC6 used in this work belongs to the collection of the Postharvest Pathology group of IRTA (Lleida, Spain) and was isolated in September 2010 from a latent infection in peaches harvested in Alfarràs orchards (Spain). This strain was identified by the Department of Plant Protection, INIA (Madrid, Spain) and deposited in the Spanish Culture Type Collection (CECT 21105). For long-term storage, the isolate was kept as conidial suspension in 20% glycerol at −80 °C and was grown on malt extract plates (MEA) at 20 °C in darkness for 10–12 days.

### 2.2. DNA and RNA Isolation

Mycelium was grown on cellophane sheets on malt extract agar and high-molecular-weight DNA was extracted from freeze-dried mycelium using a Puregene DNA purification kit from Qiagen (Venlo, The Netherlands) using manufacturer’s recommendations. Briefly, mycelium was treated with cell lysis solution, proteinase K and protein precipitation solution. DNA was precipitated with isopropanol, and after washing with 70% ethanol and drying, it was dissolved in TE buffer amended with RNase. To increase purity, DNA was cleaned by a salt:chloroform wash (Pacific Biosciences shared protocol).

RNA-Seq libraries were generated from pools of RNA isolated from in vitro and in vivo samples. In vitro samples included 2-day-old mycelia grown in liquid minimal media and spores collected from a 10-day-old culture grown on potato dextrose agar supplemented with blended tomato leaves. In vivo samples included infected peaches and nectarines (with or without wounding before inoculation) and infected peach leaves. Total RNA was extracted as described by Vilanova et al. [25] based on the conventional CTAB method. RNA quality was checked with an Agilent 2100 system.

### 2.3. Sequencing and de Novo Assembly of the Genome

*M. fructicola* DNA was sequenced using PacBio sequencing technology on a Sequel instrument by Keygene N.V. (Wageningen, the Netherlands). De novo assembly was performed using HGAP [26] and CANU [27] using default settings. Results from the assembly were combined with Quickmerge [28], and two steps of corrections were run with Arrow. A visual inspection was done to manually correct contigs that were erroneously merged. Completeness of the assembly was assessed using the Benchmarking Universal Single-Copy Orthologs (BUSCO) tool [29]. The transcriptome of *M. fructicola* (in vitro and infected plant samples) was sequenced using paired-end libraries with Illumina HiSeq-^TM^ 4000 (read lengths of 2 × 150 bp) at the Beijing Genome Institute (BGI, Hong Kong, China). The number of reads for the in vitro RNA libraries and the infected plant samples was 79.5 and 66.3 M, respectively. More than 96.5% of the base calls exceeded a quality score of 20. 

### 2.4. Genome Annotation and Expression Quantification

After genome assembly, gene models were predicted following the FunGAP pipeline [30], which uses MAKER [31], AUGUSTUS [32] and BRAKER [33] supported by evidence from RNA-Seq data from in vitro and infected plant samples. To obtain more accurate gene prediction, we aligned the *M. fructicola* gene models to the manually curated *B. cinerea* genome [34,35] and to all fungal proteins available in the SwissProt database. *M. fructicola* predicted proteins were manually curated when needed and functionally annotated using the pipeline funannotate [36]. RNA-Seq reads were mapped on the *M. fructicola* CPMC6 genome using HISAT v.2.0.3-beta. Read counts for each gene model were normalized to the total amount of mapped reads in the sample (Counts Per Million). Since RNA samples consisted of mixtures of different tissues and cell types, there were no biological replicates and the results could not be statistically analysed. 

### 2.5. Secondary Metabolite Gene Cluster Analysis

The prediction of secondary metabolite gene clusters in the genome of *M. fructicola* was performed as described by Valero-Jiménez et al. [37]. Gene clusters predicted to be involved in the biosynthesis of secondary metabolites (BGCs) were identified using AntiSMASH, version 4.1.0 using default settings [38]. The BGCs were analysed using BIG-SCAPE version 2018.10.05 [39] with a cutoff of 0.65 and the MIBiG parameter, which contained annotated secondary metabolites clusters (MIBiG repository version 1.4 [40]). A manual filtration was done to remove clusters containing mainly enzymes with low bitscore. Finally, protein–protein BLAST (version 2.7.1+) was used to calculate identity scores of *M. fructicola* proteins against the proteins of *B. cinerea*. 

### 2.6. Secretome and Effector Prediction

Genes encoding secreted proteins in the *M. fructicola* genome were identified using several tools. Signal-P v4.1 [41] was used to screen for a signal peptide, followed by TMHMM v.2.0 [42] to identify putative transmembrane domains. Proteins that lacked a signal peptide or that had a transmembrane domain (a single domain in the N-terminal 60 residues was allowed) were discarded. TargetP was used to predict protein localization [43]. CAZy enzymes were annotated using dbCAN2 meta server [44]. Effectors were predicted using the EffectorP tool versions 1 and 2 [45,46]. 

### 2.7. Characterization of Candidate Effector Proteins

#### 2.7.1. Amplification of Candidate Effector Genes

Primers were designed to cover the coding sequence (CDS) of the mature candidate effector, but excluding the N-terminal signal peptide. Primers were designed containing a six-nucleotides extension upstream of the restriction enzyme recognition site to ensure efficient cleavage (Appendix A). Amplification of candidate effector cDNA was performed in a total volume of 50 µL containing 0.2 µM of both forward and reverse primer, 0.2 mM of each dNTP, 1x PFU buffer (Promega, Leiden, The Netherlands), 1.5 U PFU polymerase (Promega), and 2 µL of template (80 ng cDNA from *M. fructicola* mycelium). PCR program was 95 °C for 2 min; 10 cycles of 95 °C for 30 s, 58 °C for 30 s and 72 °C for 2 min and 25 cycles with an annealing temperature of 60 °C instead of 58 °C, ending with a final step at 72 °C for 5 min. PCR products were visualized in 1–3% agarose gels.

#### 2.7.2. Ligation of Gene Constructs

PCR products and plasmids were digested using Fast digest enzymes (Thermo Fisher Scientific, Breda, The Netherlands), using either *Xho*I, *Pst*I, *Sal*I, or *Mph*1103I (*Nsi*I) in 1x Fast digest^®^ buffer (Thermo Fisher) at 37 °C, using manufacturer’s recommendations. Digested PCR products were ligated with T4 DNA Ligase (Promega) using a 5:1 molar ratio of insert PCR product:vector and incubated at 14 °C overnight. The ligated plasmid was transformed in ultra-competent DH5α *Escherichia coli* cells. Plasmids were isolated using the QIAprep Spin Miniprep kit (Qiagen) according to the manufacturer’s protocol.

#### 2.7.3. Transient Effector Gene Expression in *Nicotiana benthamiana*

Fifty ng of plasmid was added to 50 µL of electro-competent *Agrobacterium tumefaciens* cells (strain GV3101). A single transformed colony was grown in 15 mL YEB medium supplemented with 20 µM acetosyringone, kanamycin, gentamicin and rifampicin at 28 °C. The culture was centrifuged and suspended in MMAi buffer at OD600 = 0.8. After 2 h incubation, cells were infiltrated using a needleless 1 mL syringe (Omnifix^®^-F, Braun, Oss, The Netherlands) in leaves of 4–6 weeks old *N. benthamiana* plants. Pictures of responses in *N. benthamiana* leaves were taken 5 or 6 days postinfiltration.

## 3. Results

### 3.1. Sequence Assembly and Annotation

*M. fructicola* was sequenced using long read single molecule technology at 92 X coverage. The size of the assembled genome was 42.98 Mb with 98 contigs and a contig N50 of 988 Kb (Table 1). In terms of contig numbers, this assembly ranges in between the *M. fructicola* genomes published by De Miccolis Angelini et al. [22] with 20 contigs, by Rivera et al. [23] with 2643 contigs and by Kohn [47] with 3535 contigs (Table 1).

BUSCO analysis indicated a level of completeness of 98.7% with only a few fragmented (0.7%) or missing (0.5%) BUSCO orthologs. The PacBio assembly process introduced more than 100 indels, which were checked individually and manually fixed. After the prediction of gene models using the FunGAP pipeline, the proteome of *M. fructicola* was entirely manually curated to fix erroneous methionine start codon predictions, to remove pseudogenes and to remove gene models predicted in transposons and repetitive regions. During this process, a total of 2,857 predicted genes were deleted, resulting in a curated *M. fructicola* proteome, which contains 10,086 proteins (Figure 1).

### 3.2. Secondary Metabolites 

Fungi produce a spectrum of secondary metabolites (SMs) that are synthesized by proteins encoded in genes that are commonly arranged in biosynthetic gene clusters (BGCs) on the genome [48]. SMs play important roles in the development of fungi, their adaptation to different environments and interactions with other microbes. A total of 31 BGCs were detected and classified based on the type of SM they synthesize (Figure 2). *M. fructicola* predominantly contains BGCs involved in the production of polyketides (Type I PKS, 16), non-ribosomal peptides (NRPS, 8) and terpenes (TS, 4).

Compared with the related species *B. cinerea*, for which SM BGCs have been well studied, *M. fructicola* contains two orthologous clusters involved in the production of melanin, the brown pigment present in sclerotia and conidia. The inability to produce this metabolite by *M. fructicola* yields albino conidia and sclerotia and reduces the virulence in peaches [49]. The melanin biosynthetic pathway in *B. cinerea* contains two separate PKS key enzymes (PKS12 and PKS13) as well as enzymes acting downstream (Bcbrn2, Bcscd1 and Bcbrn1) to synthesize melanin [50]. *M. fructicola* has both PKS12 and PKS13 clusters and like in *B. cinerea* and other Sclerotiniacea species, they are not physically clustered but they both use the same set of enzymes to produce melanin. Mfpks13 is clustered with Mfscd1 and Mfbrn2, while Mfpks12 is clustered with the transcription factor Mfsmr1. Genes encoding key enzymes from both clusters show 88% similarity to orthologs in *B. cinerea*.

Moreover, *M. fructicola* contains genes for the synthesis of the phytotoxic polyketide botcinic acid, however the genes are organized differently as compared to *B. cinerea*. The botcinic acid BGC in *B. cinerea* is located at the start of Chr1 and contains 13 genes, including the BcBOA6 and BcBOA9 genes that encode the two key PKS enzymes [17,35]. The *M. fructicola* genome contains orthologs to all 13 BOA genes, however, divided over two separate genomic locations that are not telomeric as in *B. cinerea*. In *M. fructicola*, the genes BOA1 and BOA2 are together in a location on contig MFRU002, while the genes BOA3-13 are clustered in a different location on contig MFRU064. The BOA gene configuration in *M. fructicola* very much resembles that in *Sclerotinia sclerotiorum* [37], which prompted us to examine this in some more detail (Figure 3). The loci that carry the BOA1 and BOA2 genes are located in syntenic regions in *M. fructicola* (MFRU002) and *S. sclerotiorum* (Chr5), as flanking genes on either side of the cluster are orthologous. In *B. cinerea*, *B. aclada* and *B. porri*, however, the orthologs of the flanking genes of BOA1 and BOA2 in *M. fructicola* and *S. sclerotiorum* are not located in syntenic regions (Figure 3A). In addition, for the BOA3-BOA13 cluster, flanking genes on either side of the cluster are orthologous between *M. fructicola* (MFRU064) and *S. sclerotiorum* (Chr15). By contrast, *B. cinerea*, *B. aclada* and *B. porri*, contain a region that is syntenic to these flanking regions, but lack the BOA cluster (Figure 3B).

### 3.3. Secreted Proteins

From 10,458 proteins, there were 855 proteins predicted to be secreted, which corresponds to 8% of the proteome. This result is concordant with the proportion of secreted proteins in other members of the *Sclerotiniaceae* family [35,37]. Plant pathogenic fungi use secreted carbohydrate active enzymes (CAZymes) mainly to break down plant tissue to acquire nutrients and establish the infection or they secrete effector proteins to manipulate the defence responses of their host plants. In *M. fructicola*, 14% of the secreted proteins correspond to CAZymes (122) and 16% correspond to predicted effector proteins (134). We analysed in depth the proteins related to the degradation of plant cell wall carbohydrates. CAZymes are subdivided depending on their activity as glycoside hydrolases (GH), glycosyl transferases (GT), polysaccharide lyases (PL) and carbohydrate esterases (CE). In *M. fructicola*, more than half of the secreted CAZymes (81) belong to the GH category (Figure 4). 

Specifically, the CAZY families with the largest number of members are the polygalacturonases (GH28; 18 genes), xyloglucanases (GH16; 11 genes), cutinases (CE5; 7 genes) and pectin/pectate lyases (PL1 and PL3; 6 genes). Classification of CAZymes based on the preferred substrate showed that the pectin degrading capacity of *M. fructicola* is larger compared to enzymes involved in hemicellulose or cellulose degradation (Table 2).

### 3.4. Effector Proteins

In order to prioritize predicted effector genes for functional characterization, we filtered out genes that are less likely to be virulence determinants during infection of stone fruit. From the 134 predicted effector genes, 65 genes were removed because of low expression (<1 CPM) in the infected plant samples and 17 genes were removed because they contained a protein domain with known enzymatic activity. The remaining 52 effector genes were divided in three different groups based on their gene expression in infected plant samples and in vitro (Figure 5). A total of 17 effector genes showed similar expression levels in both conditions (ratio P/V between 0.5 and 2.0), while 10 genes were higher expressed in planta (ratio P/V > 2.0) and 25 were higher expressed in vitro (ratio P/V < 0.5). 

The 52 effector genes were also classified based on attributes considered to be important for effectors (Figure 6): 37 had a size smaller than 200 amino acids, 27 contained more than 4 cysteine residues, and 6 had a Pfam domain unrelated to enzymatic activity. The majority of these genes (48 of 52) had homologs in other *Sclerotiniaceae* species.

The list of 52 candidate effector proteins was refined to 33 (Table 3) by removing proteins that were predicted by only one of the two versions of EffectorP [45,46]. 

### 3.5. Characterization of M. fructicola Candidate Effectors

From the previous list, we selected five effector proteins to be tested for their cell death-inducing capacity (highlighted bold in Table 3), as well as the *M. fructicola* NEP-like protein MFRU_030g00190, which is the ortholog of BcNep2, previously shown to be a strong necrotizing effector [51]. For each of the five genes, the mature protein coding sequences were cloned into a binary vector containing the CaMV 35S promoter for transient expression and the tomato PR1a signal peptide sequence suitable for secretion into the plant apoplast. Constructs were made with and without a C-terminal Myc-tag and introduced into *A. tumefaciens*. The effector proteins were transiently expressed by agroinfiltration in *N. benthamiana* leaves and symptoms were monitored over time. MfNep2 and two candidate effectors of unknown function were able to induce cell death in infiltrated areas (Figure 7). 

## 4. Discussion

*M. fructicola* is considered the most economically damaging *Monilinia* spp. in stone fruit since it is over the past decade displacing endemic European *Monilinia* spp. such as *M. fructigena* and *M. laxa*. Its invasive success may be promoted by its capacity to produce higher amounts of conidia [4], as compared to other *Monilinia* species. In the last 2 years, several laboratories have initiated efforts to generate genome data for *Monilinia* species to gain more knowledge about this fungal genus [22,23,24,47].

In the present study, PacBio sequencing technology was used to assemble the genome of *M. fructicola* strain CMPC6, followed by a rigorous structural annotation based on a manual curation of the proteome. The PacBio assembly process yielded a lower number of contigs compared to the assemblies of two other *M. fructicola* strains generated with Illumina [23,24], which had an automated genome annotation of lower quality. An improved draft genome of *M. fructicola* strain Mfrc123 was generated using a hybrid assembly strategy, which combined both Illumina and PacBio technologies [22]. The genome assembly of strain Mfrc123 was clearly improved in terms of contiguity and contained 12,118 genes, however, for unknown reasons, it had a lower BUSCO score than one might expect for a near-chromosome-size assembly. The genome assembly of strain CPMC6 sequenced in this study had similar numbers of genes based on automated prediction tools. However, the number of trustworthy, high-quality gene models was reduced to 10,086 by a manual curation effort. The 2857 gene models that were deleted contained several types of errors caused by incorrect translation start site selection, improper prediction of splice junction and pseudogenes, among others. The resulting number of 10,086 genes is consistent with other *Monilinia* species such as *M. laxa* (9567 genes [52]) and *M. fructigena* (10,502 genes [53]), however, both latter genomes were also not manually curated. Other *Sclerotiniaceae* family members from the genera *Botrytis*, *Sclerotinia* and *Sclerotium* for which manually curated proteomes have been generated contain between 11,107 and 11,963 genes [35,37,54], at least 1000 genes more than *M. fructicola* strain CMPC6. The accurate structural and functional annotation of the *M. fructicola* genome will be a key asset in transcriptome studies of infected plant material and will enable functional comparisons of physiological processes in the pathogen during infection of different host species and tissues (flowers, fruit, leaves or twigs). 

The *M. fructicola* genome contains a set of 31 BGCs encoding enzymes involved in the synthesis of secondary metabolites, substantially fewer than *Botrytis* species that have around 40–50 clusters [37]. The biosynthetic cluster for botcinic acid (BOA) in *B. cinerea* contains 13 genes at the start of Chr1, whereas the BOA genes in *S. sclerotiorum* are divided over two chromosomal locations on Chr5 (genes BOA1 and BOA2) and Chr15 (BOA3-13) [37]. Other *Botrytis* species also contain a cluster of 13 BOA genes, however, in different genomic locations and phylogenetic analysis, it was shown that BOA clusters in *B. cinerea* and *B. sinoallii* were acquired by horizontal transfer from an ancestral taxon closely related to *B. aclada* or *B. porri* [37]. The analysis of the *M. fructicola* genome shows that the BOA genes are dispersed over two chromosomal clusters, similar to *S. sclerotiorum* and distinct from the *Botrytis* species. The configuration of both clusters is syntenic between *M. fructicola* and *S. sclerotiorum*, both for the BOA genes and the genes flanking the clusters, but distinct from the configuration in *Botrytis* spp. The configuration of BOA genes in *M. fructicola* and *S. sclerotiorum* probably represents the ancestral configuration in the *Sclerotiniaceae*. Clustering of all 13 BOA genes is a specific feature of *Botrytis* spp., and probably resulted from a physical reassorting of the ancestral clusters during the divergence of the *Botrytis* lineage from other *Sclerotiniaceae*. Within the genus *Botrytis*, the BOA cluster has subsequently migrated to distinct chromosomal locations by independent rearrangement events [37].

One characteristic of pathogens with a necrotrophic lifestyle is their ability to decompose complex plant carbohydrates by the secretion of CAZymes, which release sugar monomers that serve as carbon sources. The *B. cinerea* and *S. sclerotiorum* genomes contain 367 and 346 genes encoding putative CAZymes, respectively, which include 106 and 118 associated with plant cell wall (PCW) degradation [55]. *M. fructicola* contains 90 genes encoding PCW degrading enzymes, fewer than the above close relatives but almost double the number in the unrelated postharvest pathogen *P. digitatum* [56]. *M. fructicola* has fewer genes encoding hemicellulose degrading enzymes than *S. sclerotiorum* or *B. cinerea* and this could be related with its preference to infect fruit (predominantly rich in pectins) rather than vegetative tissues, which contain more hemicelluloses than fruit. This is consistent with the observation that *M. fructicola* grows better on pectin-based media [57] than on xylan and cellulose, with the highest growth rate in apple pectin medium [58]. 

The importance of the capacity to degrade pectin (the major carbohydrate in *Prunus* fruit) for *M. fructicola* could be reflected in the number of polygalacturonases (PGs) belonging to family GH28. Similar observations were reported for the tomato-*Rhizopus stolonifer* interaction in which the majority of differentially expressed genes encoding PCW degrading enzymes contribute to pectin degradation [59]. All *Sclerotinia* and *Botrytis* species analysed contain 17–19 GH28 genes [37,60] and also *M. fructicola* contains 18 GH28 genes. In *M. laxa*, only one PG was identified in the exoproteome [61] but two PG genes were transcriptionally induced in infected nectarine fruit [62]. In *B. cinerea*, the deletion of either BcPG1 or BcPG2, separately, resulted in virulence reduction on several host plants [63,64]. Functional molecular-genetic information about the importance of PCW degrading enzymes in *Monilinia* species is only available for the *MfCut1* gene. Overexpression of this gene resulted in higher virulence of the pathogen [20]. The structural and functional annotation performed in this study, in combination with detailed transcriptome analyses of infected plant tissue will help characterize the genes encoding PCW-degrading enzymes and prioritize genes for knockout studies. 

It is well established that fungal plant pathogens secrete many effector proteins, which are very important for pathogenesis in fungi with biotrophic, hemi-biotrophic and necrotrophic lifestyles. The present study aimed to identify the *M. fructicola* effector repertoire and to focus on cell death-inducing proteins, which can contribute to virulence. To our knowledge, there are no studies yet on the effector repertoire of *Monilinia* species. Studies in *B. cinerea* and *B. elliptica* focused on the capacity of necrosis and ethylene-inducing proteins (NLPs) to induce host cell death and on the role of NLPs in pathogenicity. However, in both fungi, single knockout mutants in Nep1 and Nep2 were not affected in virulence [16,65]. Another *B. cinerea* cell death-inducing effector studied was a cerato-platanin protein (Bcspl1) encoded by the *Bcspl*1 gene, which was highly expressed in infected plant samples [18]. *Bcspl*1 knockout mutants were generated and displayed reduced virulence in different hosts [18]. In the case of *M. fructicola*, the accurate structural and functional annotation of the genome enabled us to define a list of secreted effector proteins that can be analysed for functions in virulence. In this study, we showed that Mfru030g00580 and Mfru020g05260 proteins were able to induce a cell-death response in *N. benthamiana* leaves. The Mfru030g00580 gene has homologs in 17 *Botrytis* species but neither in *S. sclerotiorum* nor in *S. cepivorum,* while the Mfru020g05260 gene has homologs in *Sclerotinia* and *Botrytis* species and in *M. fructigena*. The function of both genes in these other *Sclerotiniaceae* is unknown. The roles of effectors in *M. fructicola* could be diverse, either in suppressing host immune responses during the early, biotrophic phase of the infection or in inducing plant cell death, possibly by host-specific programmed cell death induction in the necrotrophic phase [10]. The experiments in this study on cell death induction in *N. benthamiana* will need to be expanded to stone fruit tissues, however, heterologous protein expression by agroinfiltration in *Prunus* species remains to be developed. An alternative strategy for heterologous protein production might be via recombinant plant viruses. Tobacco Rattle Virus can replicate in *Prunus* [66], and it may be feasible to use modified TRV constructs for expressing *M. fructicola* effector proteins in *Prunus* leaves and thereby test their biological activity. Besides testing the cell death-inducing activity of *M. fructicola* effector proteins, their role in virulence can be analysed with the use of knockout mutants using a CRISPR-Cas9-based transformation protocol recently described for *Botrytis cinerea* [67]. 

The high quality of the *M. fructicola* CMPC6 genome assembly and its structural and functional annotation will enable to gain deeper knowledge on the genes involved in virulence and provides valuable information for studies on the genome biology and evolution of the genus *Monilinia* and the family *Sclerotiniaceae*, which contain many dozens of phyto-pathogenic fungi of great economical relevance worldwide.

## Figures and Tables

**Figure 1 genes-12-00568-f001:**
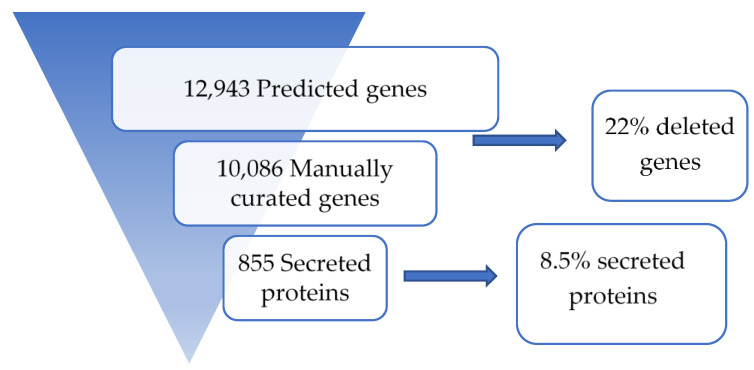
Manual curation of the annotated genome to obtain a trusted group of secreted proteins.

**Figure 2 genes-12-00568-f002:**
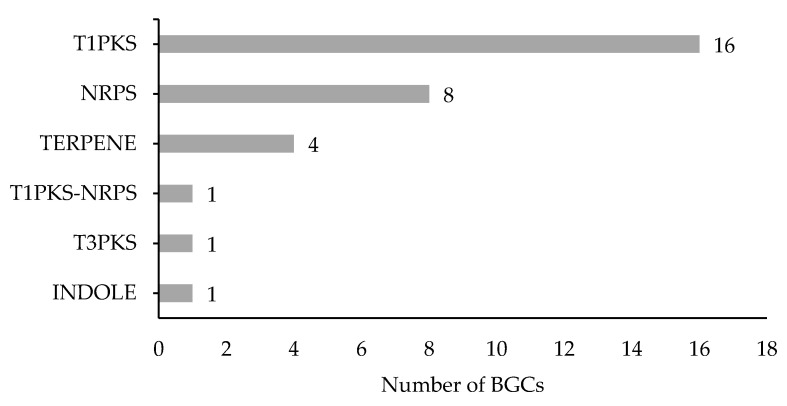
Classification of Biosynthetic Gene Clusters in *M. fructicola* based on the type of secondary metabolite they synthesize: polyketides (PKS), non-ribosomal peptides (NRPS), terpenes (TS) and alkaloids (INDOLE).

**Figure 3 genes-12-00568-f003:**
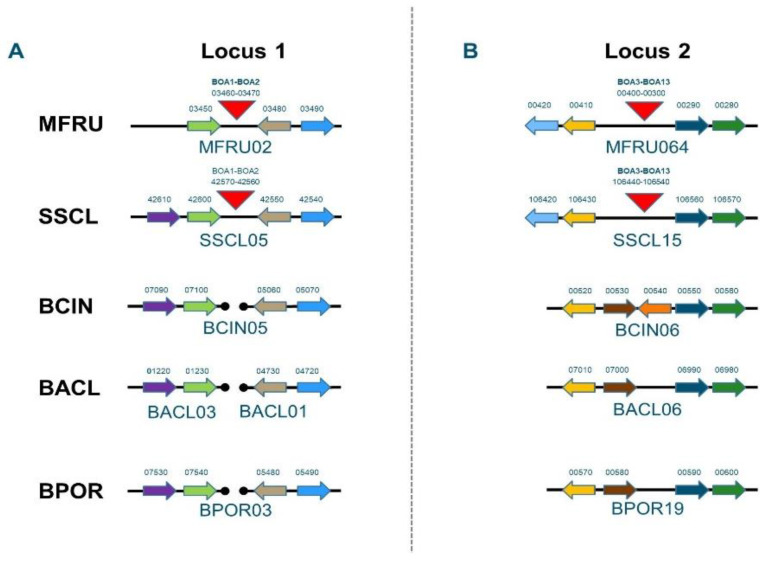
Configuration of *M. fructicola* botcinic acid (BOA) biosynthetic genes in two clusters (panels **A**,**B**), as compared to homologs in *S. sclerotiorum*, *B. cinerea*, *B. aclada* and *B. porri*. Contig numbers in the five species are provided underneath the locus. Orthologous genes are in identical colours. Gene numbers within the contig are provided above the gene, with the arrow indicating the direction of transcription. The triangular red blocks represent the BOA cluster. Synteny breaks are shown by interrupted lines with dots marking the break.

**Figure 4 genes-12-00568-f004:**
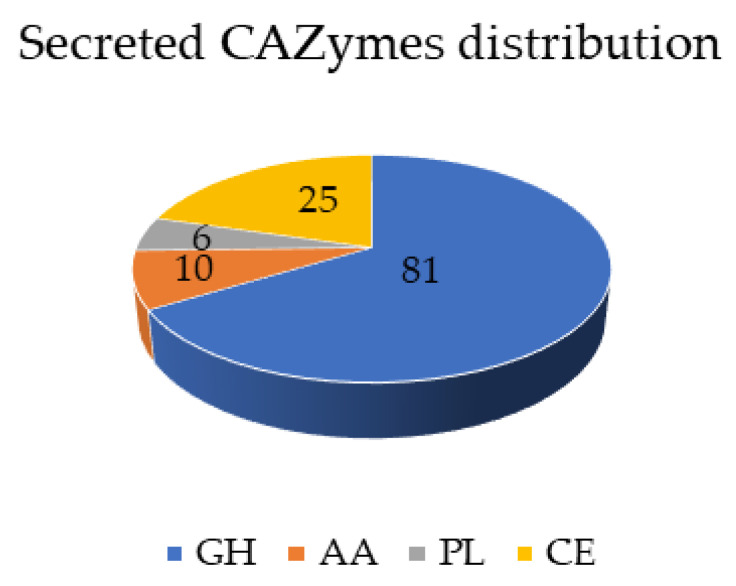
Distribution of secreted carbohydrate active enzymes (CAZymes) in *M. fructicola* genome. Enzymes were categorized as glycoside hydrolases (GH), polysaccharide lyases (PL), carbohydrate esterases (CE), and auxiliary activity (AA). Numbers in the pie chart indicate the total gene numbers in each category.

**Figure 5 genes-12-00568-f005:**
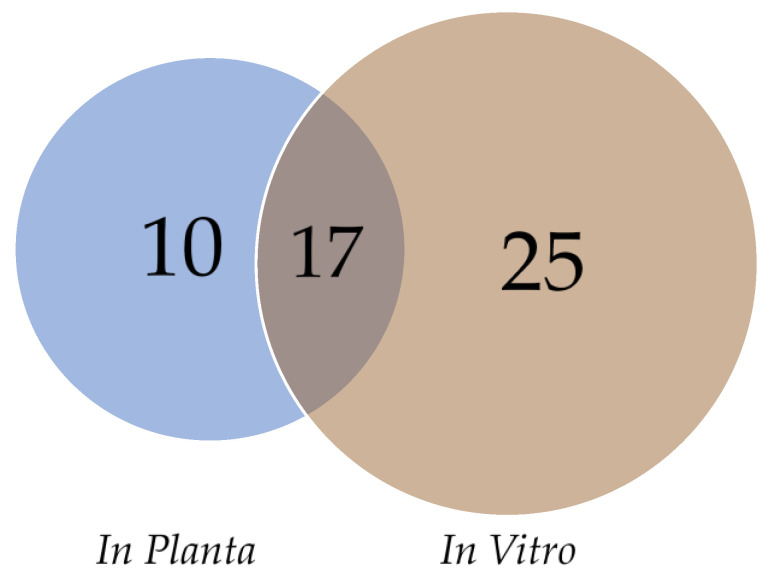
Venn diagram for the effector genes differentially expressed depending on the conditions (in vitro or infected plant samples).

**Figure 6 genes-12-00568-f006:**
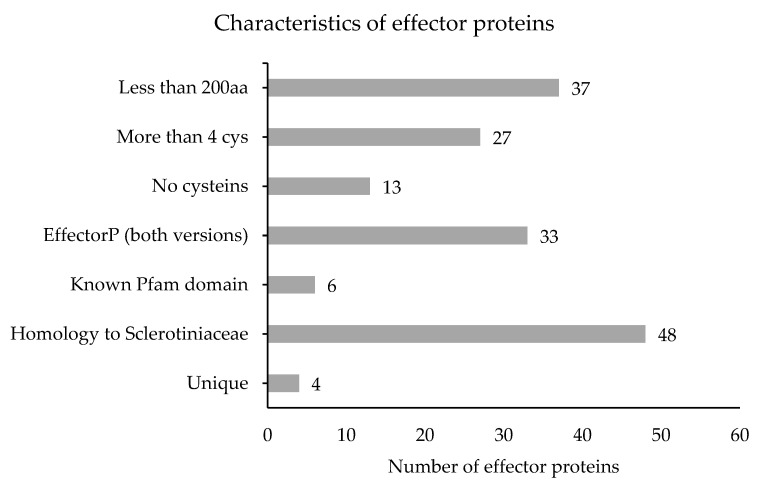
Characteristics of selected effector proteins.

**Figure 7 genes-12-00568-f007:**
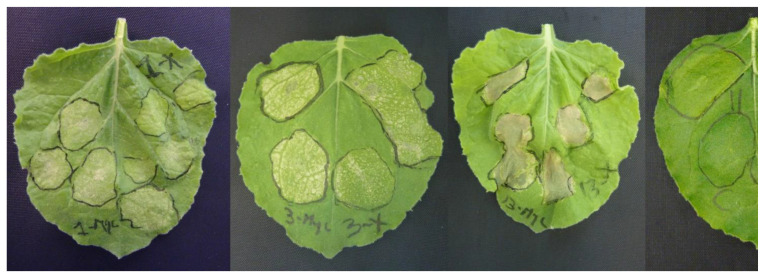
Transient expression of *M. fructicola* candidate effector genes inducing cell death in *N. benthamiana*. From left to right: MFRU_002g05260, MFRU_030g00580, MfNep2 and empty-vector control. The left leaf halves were infiltrated with a construct carrying a Myc-tagged effector coding sequence, while right leaf halves were infiltrated with untagged constructs. Constructs of candidate effector genes MFRU_012g01440, MFRU_034g00500 and MFRU_048g00370 looked similar to the empty-vector control. Pictures were taken 6 days postinfiltration.

**Table 1 genes-12-00568-t001:** Assembly features and gene prediction information of *M. fructicola* genomes.

Isolate	Genome Size (Mb)	Number of Contigs	Contig N50 (Kb)	Contig L50 (bp)	GC (%) Fraction	BUSCO Complete (Partial)	Predicted Genes	Reference
CMPC6	42.95	98	988	14	41.5	98.7 (99.5)	10,086	This work
Mfrc123	44.05	20	2592	7	40.8	88 (98)	13,749	De Miccolis Angelini et al. (2019)
BR-32	42.82	2643	62	205	41.7	>97%	NA ^1^	Rivera et al. (2018)
LMK 125	44.68	3535	37	353	40.1	NA ^1^	NA ^1^	Kohn (2017)

^1^ NA (not available).

**Table 2 genes-12-00568-t002:** Comparison of numbers of genes encoding plant cell wall degrading enzymes (PCWDE) and fungal and plant cell wall degrading enzymes (FPCWDE) in the genomes of *Sclerotiniaceae* species and other ascomycetes. The substrate preferences of the enzymes are provided for C (cellulose), H (hemicellulose), HP (hemicellulose or pectin side chains) and P (pectin).

Fungal Species	PCWDE						FPCWDE	References
	Total	C	H	HP	P	H/HP/P		
*M. fructicola*	90	18	27	12	33	72	22	This work
*S. sclerotiorum*	106	20	40	13	33	86	32	Anselem et al., 2010
*B. aclada*	89	17	27	13	32	72	31	Valero-Jiménez et al., 2020
*B. cinerea* T4	118	18	41	15	44	100	36	Anselem et al., 2010
*Neurospora crassa*	78	26	31	13	8	52	22	Anselem et al., 2010
*Penicillium digitatum*	49	NA ^1^	NA ^1^	NA ^1^	NA ^1^	NA ^1^	NA ^1^	Marcet-Houben et al., 2012

^1^ NA (not available).

**Table 3 genes-12-00568-t003:** Annotations of candidate effectors and their expression in vitro and in infected plant samples.

Effector	Protein Size (aa)	# Cys	*In Planta* CPM	*In Vitro* CPM	Ratio P/V	PFAM Domain	Homology to Other Species
MFRU_001g01030	232	13	14.8	123	0.12		*Botrytis*, *Sclerotinia*
MFRU_001g04320	95	0	197	130	1.5		Many fungi
MFRU_001g04590	122	6	34.3	84	0.4		Many fungi
MFRU_001g05460	148	8	16.7	154	0.11		*Botrytis, Sclerotinia, Chaetomium, Rutstroemia*
MFRU_002g00070	246	0	9.8	0.6	15.2		*Botrytis*
MFRU_002g02190	94	10	150	0.8	181		*Aspergillus, Botrytis, Sclerotinia*
MFRU_002g03250	203	0	50	76	0.7		Many fungi
**MFRU_002g05260 ^1^**	**221**	**10**	7.3	8.0	0.9	**PF05730** (**CFEM domain**)	***Botrytis, Sclerotinia***
MFRU_003g02920	190	6	2.1	3.1	0.7		*Periconia, Sclerotinia*
MFRU_003g05140	129	2	2.4	16.8	0.14		*Botrytis, Sclerotinia*
MFRU_004g01490	109	2	11.7	45	0.3		None
MFRU_004g01570	180	4	1.4	1.3	1.0		None
MFRU_004g02710	105	6	2.8	1.3	2.1		*Aspergillus, Cladophora, Sclerotinia*
MFRU_004g02780	87	0	0.9	0.6	1.4		None
MFRU_005g03220	83	7	1	0.9	1.1		*Botrytis, Sclerotinia*
MFRU_008g03050	113	7	3.1	611	0.01		*Botrytis, Sclerotinia*
MFRU_008g03430	204	0	3.3	8.9	0.4		*Pochonia, Purpureocillium, Sclerotinia*
MFRU_009g00790	177	2	0.7	0.8	1.0		Many fungi
MFRU_012g00980	173	6	4.8	263	0.02		Many fungi
**MFRU_012g01440 ^1^**	**245**	**4**	25.3	325	0.08	**PF14021** (**tuberculosis necrotizing toxin**)	**Many fungi**
MFRU_012g01930	191	4	59	8.7	6.7		*Botrytis, Rutstroemia, Sclerotinia*
MFRU_014g01140	160	8	1	0.7	1.4		*Botrytis, Sclerotinia*
MFRU_014g02060	247	3	4.1	9.5	0.4	PF05630 (necrosis-inducing protein NPP1)	Many fungi
MFRU_015g00570	92	8	5.4	200	0.03		Many fungi
MFRU_017g01630	128	7	15.9	138	0.1		*Botrytis, Diplocarpon, Rutstroemia, Sclerotinia*
MFRU_022g00070	192	9	1.9	0.8	2.2		*Botrytis*
MFRU_028g01250	149	5	3.6	35.4	0.12	PF07249 (cerato-platanin)	*Many fungi*
**MFRU_030g00580 ^1^**	**160**	**6**	**135**	**14.9**	**9.1**		***Botrytis, Monilinia***
**MFRU_034g00500 ^1^**	**82**	**8**	**201**	**52.7**	**3.8**		***Botrytis***
MFRU_035g00290	167	8	10.9	8.7	1.2		*Botrytis, Sclerotinia*
MFRU_036g00390	263	18	6.7	49.7	0.13		*None*
**MFRU_048g00370 ^1^**	**147**	**9**	**4.4**	**1.7**	**2.6**		***Many fungi***
MFRU_062g00230	126	0	1.0	0.8	1.2		*Botrytis, Monilinia*

^1^ genes tested for cell death induction by transient expression in *N. benthamiana*.

## Data Availability

This Whole Genome Shotgun project has been deposited at DDBJ/ENA/GenBank under the accession number RKRL00000000. The version described in this paper is version RKRL01000000. The project has been deposited in GenBank under the Bioproject number PRJNA503180. The Biosamples related to this project have accession number SAMN10354043. The raw PacBio genomic read data are deposited under accession number SRR8146337.

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
