# Peer review of "Deciphering the Monilinia fructicola Genome to Discover Effector Genes Possibly Involved in Virulence"

_genes, 2021, doi:10.3390/genes12040568_

Round 1

Reviewer 1 Report

The manuscript "Deciphering the Monilinia fructicola Genome to Discover Effector Genes Possibly Involved in Virulence" applied PacBio and Illumina RNA-Seq to assembly a genome and characterize necrosis-inducing effectors. The manuscript is well written and clearly presented. In general, the manuscript meets the scientific soundness although the scientific significance to the community of molecular plant pathology remained subjective.

One reason is that the genome of Monilinia fructicola have been published several times, and although this version shows the best BUSCO, this genome version is not the longest or any close to chromosome level. The second reason is that the effectors being tested in this study have been well-studied in other fungi. Repeated validation of their necrosis capability contributes no novelty to the field, and knowledge advance from this study may be low. Considering the impact factor of Genes at 3.8, I would suggest the editors to consider if this manuscript meets the scope and vision of the journal. However, if consider the robustness of this study, the manuscript is mostly well ready for publication.

Two suggestions will be:

  1. Provide a discussion between the pros and cons of different genomes. For example, De Miccolis Angelini et al. (2019) applied PacBio and Illumina on M. fructicola as well, why is this version higher in contigs but higher in BUSCOs? Is there any difference in terms of sequencing platforms or softwares?
  2. For the Agro-infiltration assay, the authors should infiltrate the same leaf with negative control and positive control. It is well known that the condition of tobacco leaves will affect the appearance of necrosis. Under the scenario, MFRU_002g05260 and MFRU_030g00580, could be infiltration damage. With a positive control such as INF1 or others, would be easier to convince the symptoms are indeed necrosis or HR. And the half leaf to show negative control may be unnecessary.

Author Response

Comment#1

Provide a discussion between the pros and cons of different genomes. For example, De Miccolis Angelini et al. (2019) applied PacBio and Illumina on M. fructicola as well, why is this version higher in contigs but higher in BUSCOs? Is there any difference in terms of sequencing platforms or softwares?

Response

De Miccolis Angelini et al. (2019) used PacBio and Illumina sequencing data, while we used only PacBio data. De Miccolis Angelini et al. (2019) applied different software (for hybrid assembly). Their assembly is more complete, but the quality seems to be worse based on BUSCO. This could be caused by the use of another strain, containing different repeat contents, or by the lack of post-hoc quality control of the assembly. We performed several rounds of quality control and manual splitting and merging, we do not know how rigorous quality control by De Miccholis Angelini et al. has been. We have made a small change in the manuscript (lines 368-370) which now reads “The genome assembly of strain Mfrc123 was clearly improved in terms of contiguity and contained 12,118 genes, however, for unknown reasons it had a lower BUSCO score than one might expect for a near-chromosome-size assembly.”

Comment#2

For the Agro-infiltration assay, the authors should infiltrate the same leaf with negative control and positive control. It is well known that the condition of tobacco leaves will affect the appearance of necrosis. Under the scenario, MFRU_002g05260 and MFRU_030g00580, could be infiltration damage. With a positive control such as INF1 or others, would be easier to convince the symptoms are indeed necrosis or HR. And the half leaf to show negative control may be unnecessary.

Response

Several agro-infiltrations assays were performed to test the induction of cell-death response. In Figure 7, we show that necrosis can be induced in Nicotiana benthamiana leaves by two effectors, out of the five genes that we selected. MfNep2 was used as a positive control and Agrobacterium empty-vector and buffer were used as negative controls. Constructs for expressing MFRU_002g058260 and MRU_030g00580 (both with Myc-tag or without tag) triggered clear cell death symptoms, we strongly disagree with the suggestion that these responses are due to infiltration damage. Three other constructs MFRU_012g01440, MFRU_034g00500 and MFRU_048g00370 were indistinguishable from the negative controls.

We realised that the legend was somewhat incomplete and have modified the legend as follows: “Transient expression of Monilinia fructicola candidate effector genes inducing cell-death in Nicotiana benthamiana. From left to right: MFRU_002g05260, MFRU_030g00580, MfNep2 and empty-vector control. The left leaf halves were infiltrated with a construct carrying a Myc-tagged effector coding sequence, while right leaf halves were infiltrated with untagged constructs. Constructs of candidate effector genes MFRU_012g01440, MFRU_034g00500 and MFRU_048g00370 looked similar to the empty-vector control. Pictures were taken six days post infiltration.”

Reviewer 2 Report

The article provides information on another M. fructicola high-quality genome. However, in addition to just providing the characteristics of the genome, the authors have gone a step further to perform manual curation of the gene models using RNA-seq evidence data, perform transient expression of candidate effectors in Nicotiana benthamiana and delineate the effector repertoire for M. fructicola.   The authors have conducted the relevant experiments to support their research and have provided a well-written, informative article which would definitely benefit the readers of Genes.   Just some minor comments, I would like the authors to address:   The BUSCO score for the genome completeness for M. fructicola (Rivera et al) is 97-98%. This has been mentioned in the article. Please check and add this information.   Additionally, how was the completeness of the transcriptome analyzed? BUSCO assesses transcriptome completeness as well.   Library preparation steps are missing. Which kit was used? How was the concentration or integrity of the extracted DNA/library checked (Qubit/Tapestation)?   How many reads were obtained after sequencing? How many were discarded? What was the Quality cut off used for discarding the reads? 

Author Response

Comment#1
The BUSCO score for the genome completeness for M. fructicola (Rivera et al) is 97-98%. This has been mentioned in the article. Please check and add this information.   

Response  

Apologies for this omission, the information from Rivera et al. (2018) was added in Table 1.

Comment#2

How was completeness of the transcriptome analyzed? BUSCO assesses transcriptome completeness as well. 

Response  

Our BUSCO analysis on the assembly indicates that there are 98.7% complete and 99.5% partial BUSCO genes. Those scores are excellent as compared to many other fungal genome assemblies. We don’t see the added value of performing this analysis, especially considering the possibility that some BUSCO genes that are present in the assembly might NOT be expressed in the RNA samples we analysed and could therefore be missing in the BUSCO analysis on the transcriptome. BUSCO analysis on the assembly is the most commonly performed in other papers.

Comment#3
Library preparation steps are missing. Which kit was used? How was the concentration or integrity of the extracted DNA/library checked (Qubit/Tapestation)? How many reads were obtained after sequencing? How many were discarded? What was the Quality cut off used for discarding the reads?

Response
The sequencing was outsourced to BGI Hongkong and not done in our lab. We have added the requested information provided by BGI in the Materials and Methods section (lines 121 and 132-134).